# Deletion of a putative promoter-proximal *Tnfsf11* regulatory region in mice does not alter bone mass or *Tnfsf11* expression in vivo

Ryan S. MacLeod[1,2], Mark B. Meyer[3], Jinhu Xiong[1,4], Keisha M. Cawley[1,2], Yu Liu[1,2], Melda Onal[1,5], Nancy A. Benkusky[3], Jeff D. Thostenson[1,6], J. Wesley Pike[3], Charles A. O'Brien[1,2,4,7]*

1 Center for Musculoskeletal Disease Research, University of Arkansas for Medical Sciences, Little Rock, AR, United States of America, 2 Division of Endocrinology, University of Arkansas for Medical Sciences, Little Rock, AR, United States of America, 3 Department of Biochemistry, University of Wisconsin-Madison, Madison, WI, United States of America, 4 Department of Orthopaedic Surgery, University of Arkansas for Medical Sciences, Little Rock, AR, United States of America, 5 Department of Physiology and Biophysics, University of Arkansas for Medical Sciences, Little Rock, AR, United States of America, 6 Department of Biostatistics, University of Arkansas for Medical Sciences, Little Rock, AR, United States of America, 7 Central Arkansas Veterans Healthcare System, Little Rock, AR, United States of America

* caobrien@uams.edu

**Data Availability Statement:** All relevant data are within the manuscript and its Supporting Information files.

## Abstract

The cytokine RANKL is essential for osteoclast formation during physiological and pathological bone resorption. RANKL also contributes to lymphocyte production, development of lymph nodes and mammary glands, as well as other biological activities. Transcriptional control of the *Tnfsf11* gene, which encodes RANKL, is complex and involves distant regulatory regions. Nevertheless, cell culture studies suggest that an enhancer region near the transcription start site is involved in the control of *Tnfsf11* expression by hormones such as 1,25-$(OH)_2$ vitamin $D_3$ and parathyroid hormone, as well as the sympathetic nervous system. To address the significance of this region in vivo, we deleted the sequence between -510 to -1413 bp, relative to *Tnfsf11* exon 1, from mice using CRISPR-based gene editing. MicroCT analysis of the femur and fourth lumbar vertebra of enhancer knockout mice showed no differences in bone mass compared to wild type littermates at 5 weeks and 6 months of age, suggesting no changes in osteoclast formation. RNA extracted from the tibia, fifth lumbar vertebra, thymus, and spleen at 6 months of age also showed no reduction in *Tnfsf11* mRNA abundance between these groups. However, maximal stimulation of *Tnfsf11* mRNA abundance in cultured stromal cells by PTH was reduced approximately 40% by enhancer deletion, while stimulation by 1,25-$(OH)_2$ vitamin $D_3$ was unaffected. The abundance of B and T lymphocytes in the bone marrow did not differ between genotypes. These results demonstrate that the region between -510 and -1413 does not contribute to *Tnfsf11* expression, osteoclast support, or lymphocyte production in mice under normal physiological conditions but may be involved in situations of elevated parathyroid hormone.

**Funding:** This work was supported by the National Institutes of Health (R01AR049794 and P20GM125503), the Veterans Administration (I01 BX000294), and by UAMS tobacco settlement funds to CAO, and R01AR074993 to JWP. The funders had no role in study design, data collection and analysis, decision to publish, or preparation of the manuscript.

**Competing interests:** I have read the journal's policy and the authors of this manuscript have the following competing interests: CAO owns stock in Radius Health, Inc. This does not alter our adherence to PLOS ONE policies on sharing data and materials.

## Introduction

Receptor activator of NFκB ligand (RANKL) is a TNF-family cytokine that controls the differentiation and proliferation of a variety of cell types [1]. One of its best-characterized functions is to stimulate the differentiation of bone resorbing osteoclasts from myeloid progenitors [2]. In addition, it is required for the development of lymph nodes, microfold cells in the intestine, and the final stages of mammary gland development [3,4]. RANKL is also required for optimal production of mature B and T lymphocytes [3]. Consistent with its diverse functions, the *Tnfsf11* gene, which encodes RANKL, is expressed by a broad range of cell types in response to a variety of different conditions and signals [5].

Hormones such as parathyroid hormone (PTH) and 1,25-dihydroxyvitamin $D_3$ [1,25 $(OH)_2D_3$] stimulate osteoclast formation in part via stimulation of *Tnfsf11* transcription [6]. In previous studies, we identified long-range enhancers that contribute to the response of murine *Tnfsf11* to PTH and 1,25$(OH)_2D_3$ in cells of the osteoblast lineage. Specifically, we used reporter constructs derived from bacterial artificial chromosomes (BACs), as well as chromatin immunoprecipitation coupled with next generation sequencing (ChIP-seq), to identify an enhancer located 75–76 kb upstream from the transcription start site (TSS) that mediates the response to both PTH and 1,25$(OH)_2D_3$ [7,8]. Importantly, deletion of this enhancer from the murine genome reduces *Tnfsf11* expression, decreases osteoclast number, and increases bone mass under normal physiological conditions [7,9]. We also identified a second enhancer, located 22 kb upstream from the TSS, that contributes to stimulation of *Tnfsf11* expression by PTH, and deletion of this region from the mouse genome also decreases *Tnfsf11* expression and bone resorption, and increases bone mass [10].

Together, these studies demonstrate that regulation of *Tnfsf11* transcription is complex and involves multiple distant enhancers. Nonetheless, studies by others have implicated regions more proximal to the transcriptional start site (TSS) in the control of *Tnfsf11* expression by PTH and 1,25$(OH)_2D_3$. For example, deletion of a sequence resembling a vitamin D response element (VDRE) from a position 922 bp upstream from the TSS blunts 1,25$(OH)_2D_3$-stimulation of a promoter-reporter construct in an osteoblastic cell line [11]. Interestingly, a second study implicated a sequence overlapping this VDRE in the stimulation of *Tnfsf11* expression by the sympathetic nervous system [12]. More recently, this same sequence has been shown to function together with a motif located 788 bp upstream from the TSS to mediate the response to PTH, as well as the response to parathyroid hormone related protein (PTHrP) [13]. Another study by the same group showed that stimulation of *Tnfsf11* expression by isoproterenol, a surrogate for activation of sympathetic nervous system signaling, also required the sequences at -922 and -788 [14]. However, our previous promoter-reporter studies have not detected any significant hormonal responsiveness of this region in transfection studies [7,15].

The goal of the current study was to determine whether the potential regulatory sequences located near the murine *Tnfsf11* TSS play a role in expression of this gene and thereby bone metabolism. We used CRISPR-mediated gene editing to delete the region between -510 to -1413 bp, relative to the TSS, from the murine genome. We found that mice lacking this region exhibited normal skeletal mass and architecture and that *Tnfsf11* mRNA abundance in tissues was unchanged, albeit stimulation by PTH was moderately reduced in primary cell cultures.

## Materials and methods

### Generation of *Tnfsf11* enhancer KO (eKO) mice

Single guide RNAs (sgRNAs) targeting regions -510 and -1413 bp upstream of the *Tnfsf11* TSS were identified using the GPP sgRNA Designer tool [16]. The sgRNA sequences for sg-1 (-510

site) and sg-2 (-1413 site) were individually introduced into the pX330 plasmid [17]. The sg-1 expressing pX330 and sg-2 expressing pX330 were combined in a 1:1 ratio and microinjected into C57BL/6-Balbc F1 zygotes. The following primers, which flank the deletion region, were used to identify potential eKO founders: eKO Forward 5'-GTTTAGCCTCAGTTATCCCTCC AA-3'; eKO Reverse 5'-CCACGATCTCAAAGACAGGTCA-3'; product size 261 bp for mice lacking the region between the cut sites. The sequence of the deleted allele was confirmed by TOPO TA-cloning (ThermoFisher, Grand Island, NY) of the 261 bp product and sequencing (**S1 Fig**). For genotyping experimental mice, a third primer located within the deletion region was introduced to the PCR for identifying wild type alleles: WT Reverse 5'-CCCACCCCAT TCTTTTCCTA-3'; product size 185 bp. Breeding of founder mice and all subsequent offspring was performed with C57BL/6J mice. All mice were fed lab diet 8640 (Envigo), provided water ad libitum, and were maintained on a half day (12 hour) light/dark cycle. The institutional Animal Care and Use Committee of the University of Arkansas for Medical Sciences approved all protocols involving eKO mice and their wild type (WT) littermates.

## Determination of bone mineral density (BMD)

BMD was measured using the PIXImus densitometer (GE-Lunar Corporation, Madison, WI). Software version 2.0 was used for acquisition and analysis of the images. Three different sites were measured. The femur measurement encompassed the right femur. The spine measurement was a rectangle that enclosed the 8 largest vertebrae (last 2 thoracic and 6 lumbar vertebrae). The total body measurement encompassed the whole body minus the calvarium, teeth, mandible, and majority of the tail (with the exception of the first few caudal vertebrae). Isoflurane sedation was used to keep the animals motionless during the 4 minute scan. Animal respiration, heart rate, and the righting reflex were monitored during and after sedation.

## Microcomputed tomography (microCT) analysis

MicroCT scanning was used to measure cortical and trabecular architecture of the femur and the trabecular architecture of the fourth lumbar vertebra (L4). Bones were dissected, cleaned of soft tissues, fixed in 10% Millonig's formalin overnight, and transferred gradually from 70 to 100% ethanol. Dehydrated bones were scanned using a model μCT40 scanner (Scanco Biomedical, Bruttisellen, Switzerland) to generate three-dimensional voxel images (1024 × 1024 pixels) of bone samples. A Gaussian filter (sigma = 0.8, support = 1) was used to reduce signal noise and a threshold of 220 was applied to all scans, at medium resolution ($E$ = 55 kVp, $I$ = 145 μA, integration time = 200 ms). Femurs were scanned beginning immediately distal to the third trochanter to the beginning of the distal growth plate. Cortical dimensions were determined at the diaphysis. Trabecular bone measurements at the distal femur were made on 151 slices beginning 8–10 slices away from the growth plate in order to avoid the primary spongiosa, and proceeded proximally. The fourth lumbar vertebra (L4) was scanned from the rostral growth plate to the caudal growth plate. Trabecular bone analyses were performed on contours of cross-sectional images, drawn to exclude cortical bone, as described for femoral trabecular bone. All trabecular measurements were made by drawing contours every 10–20 slices and using voxel counting for bone volume per tissue volume and sphere-filling distance transformation indices. Calibration and quality control were performed weekly using five density standards, and spatial resolution was verified monthly using a tungsten wire rod. Beam hardening correction was based on the calibration records. All nomenclature, symbols, and units adhered to guidelines in the literature [18].

## Bone marrow isolation and culture

Left femurs and tibias were removed from 3 eKO or 3 WT mice under sterile conditions. Soft tissues were removed using sterile gauze and the distal and proximal epiphyses were cut off the shafts using a scalpel blade. Alpha Minimum Essential Medium (ThermoFisher) supplemented with 10% fetal bovine serum (Atlanta Biologicals) and 1% penicillin-streptomycin-glutamine (ThermoFisher) was used to flush bone marrow stromal cells into a 12-well plate with marrow from similar genotyped mice pooled together. The cells were then reseeded at a density of $5x10^6$ cells per well of a 12 well plate (9 wells per genotype per time point). Cells were cultured until reaching 70–80% confluency. Three wells of each genotype were treated with vehicle (0.03% ethanol), $10 \times 10^{-7}$ M bovine PTH (1–34), or $1x10^{-8}$ M 1,25(OH)$_2$D$_3$ for 1, 4, 12, or 24 hours. The experiment was performed twice; once using male mice and once using female mice.

## Nucleic acid isolation and gene expression analysis

Total RNA was isolated from tissues by homogenization in Trizol Reagent (ThermoFisher) according to the manufacturer's instructions. The following tissues were used for RNA isolation: cortical bone from the tibia after flushing out the bone marrow and scraping the periosteum with a scalpel, whole L5 vertebrae, thymus, and spleen. The RNeasy Plus Mini kit (Qiagen, Germantown, MD) was used to extract mRNA from *ex vivo* bone marrow cultures according to the manufacturer's instructions. RNA quantity and 260/280 ratio were determined using a Nanodrop instrument (ThermoFisher), and RNA integrity was verified by resolution on 0.8% agarose gels. Five hundred nanograms of RNA was used to synthesize cDNA using the High-Capacity cDNA Reverse Transcription Kit (ThermoFisher) according to the manufacturer's directions. Transcript abundance in the cDNA was measured by quantitative PCR using TaqMan Universal PCR Master Mix (ThermoFisher) and Taqman assays. PCR amplification and detection were carried out on an ABI StepOnePlus Real-Time PCR system (ThermoFisher) as follows: 2-min holding stage at 50C followed by a 10-min initial denaturation at 95C, 40 cycles of amplification including denaturation at 95C for 15 sec and annealing/extension at 60C for 1 min. The following TaqMan assays from Life Technologies were used: RANKL (Mm00441908_m1), OPG (Mm00435452_m1), and the housekeeping gene ribosomal protein S2 (Mm00475529_m1). Gene expression was calculated using the comparative threshold cycle ($\Delta$Ct) method [19].

## Flow cytometry

Bone marrow cells were collected from left femurs and tibias of 5 female eKO and 5 female WT mice by removing both epiphyses and flushing out cells with PBS containing 3% FBS. The cells were then washed and incubated with anti-mouse CD16/CD32 (mouse BD Fc-block, catalog number 553141, BD Biosciences, San Jose, CA) for 5 minutes on ice. Then the samples were stained for 30 minutes on ice with the following antibodies: 5 µg/ml anti-CD3-FITC (145-2C11, BD Biosciences) for T cells, 2 µg/ml anti-CD19-APC-Cy7 (1D3, BD Biosciences) for B cells, 0.5 µg/ml anti-CD11b/Mac1-APC (M1/70, BD Biosciences) for monocyte progenitors, and 0.5 µg/ml anti-Ter119-PerCP/Cy5.5 (Ter-119, BD Biosciences) for erythroid cells. Cells were washed to remove unbound antibodies and then fixed with 4% paraformaldehyde in phosphate-buffered saline for 20 minutes before analysis using a BD FACS Aria flow cytometer (BD Biosciences). Data analysis was conducted using FlowJo Software (FlowJo, LLC, Ahsland, OR). Fluorescence Minus One (FMO) controls (BD Biosciences) guidance was followed for appropriate gating for the different cell populations.

## Quantification of circulating RANKL protein

Blood was collected by retro-orbital bleeding into a microcentrifuge tube and allowed to clot at room temperature for 2 hours. It was then centrifuged at $2000 \times g$ for 20 minutes to separate serum from cells. Soluble RANKL in blood serum was measured using a mouse Quantikine kit (R&D Systems, Minneapolis, MN) according to the manual provided by the manufacturer. A five-parameter regression formula was used to calculate the sample concentrations from standard curves using GraphPad statistical software.

## Statistical analysis

Two-way ANOVA, Student's t test, or Repeated Measures Mixed Effects models for measurements over time were used to detect statistically significant genotype or treatment effects, after determining that the data were normally distributed and exhibited equivalent variances. In some cases, Wilcoxon rank-sum tests were used in place of Student's t-test and log, square root, or rank transformations were used to obtain normally distributed data. Pairwise comparisons and contrasts of genotypes were estimated in the ANOVA models. All t tests were two-sided. Tukey or Benjamini-Hochberg corrections were used for multiple comparison adjustments for each family of tests of pairwise comparisons and contrasts. Statistical analyses were performed using GraphPad Prism 8 or SAS version 9.4.

## Results

We have previously identified functional enhancers located 75–76 kb and 22 kb upstream from the murine *Tnfsf11* TSS [7,8,10]. Consistent with the functional importance of these sites, the sequences of these regions are highly conserved between mice and humans [7]. In contrast, the region nearer the TSS, which has been implicated in the response to hormones and the sympathetic nervous system, located approximately -700 to -1000 bp upstream of the TSS, does not display significant evolutionary conservation (**Fig 1A**). Moreover, our previous in vitro studies demonstrated that promoter-reporter constructs containing up to 2 kb of sequence upstream from the TSS were unresponsive to either PTH or $1,25(OH)_2D_3$ [7]. This lack of responsiveness, coupled with the lack of species conservation, suggests that this region is not required for hormonal control of the *Tnfsf11* transcription. To directly address this question in vivo and in the context of the endogenous *Tnfsf11* gene, we deleted a DNA fragment from -510 to -1413 bp, relative to the TSS of the *Tnfsf11* gene, from the mouse genome (**Fig 1B**). This region contains the putative VDRE, CRE, and NFATc1 binding sites identified by others [11–14]. To perform the deletion, plasmids encoding Cas9 and sgRNAs flanking the putative binding sites were injected into murine zygotes and offspring with the desired deletion were identified by PCR and DNA sequencing (**Figs 1B and S1**).

We selected one founder mouse lacking the region between -510 and -1413, indicated by the arrows in Fig 1B, to establish a colony of mice for skeletal analysis. Mice with homozygous deletion of the -510 and -1413 region were designated enhancer knockout (eKO) and wild type (WT) littermates were used as controls. Serial analysis of bone mineral density (BMD) between 8 and 25 weeks of age revealed no differences between genotypes in either male or female mice at any of the sites examined (**Fig 2A and 2B**). While there was no difference in body weight in female mice, eKO males weighed slightly less than controls at 16 and 25 weeks (**Fig 2C and 2D**). Consistent with the latter, male eKO mice exhibited slightly lower lean and fat mass but no change in femur length (**S2 Fig**).

To determine if eKO mice exhibited differences in skeletal architecture and bone volume, we performed micro-computed tomography (micro-CT) analysis on right femurs and fourth lumbar vertebrae (L4) of mice of both sexes at 5 weeks and 6 months of age. Femoral

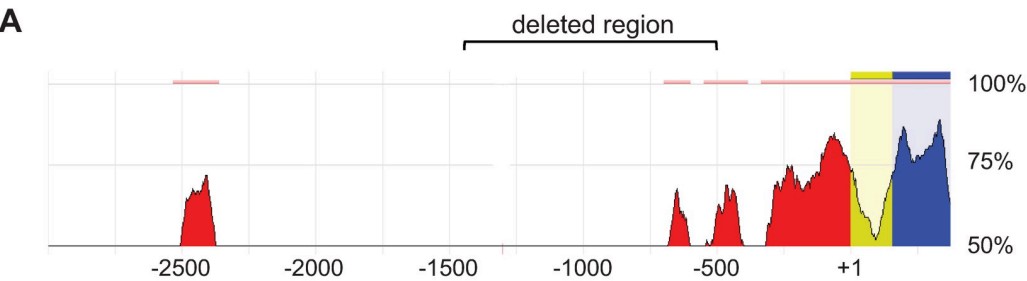

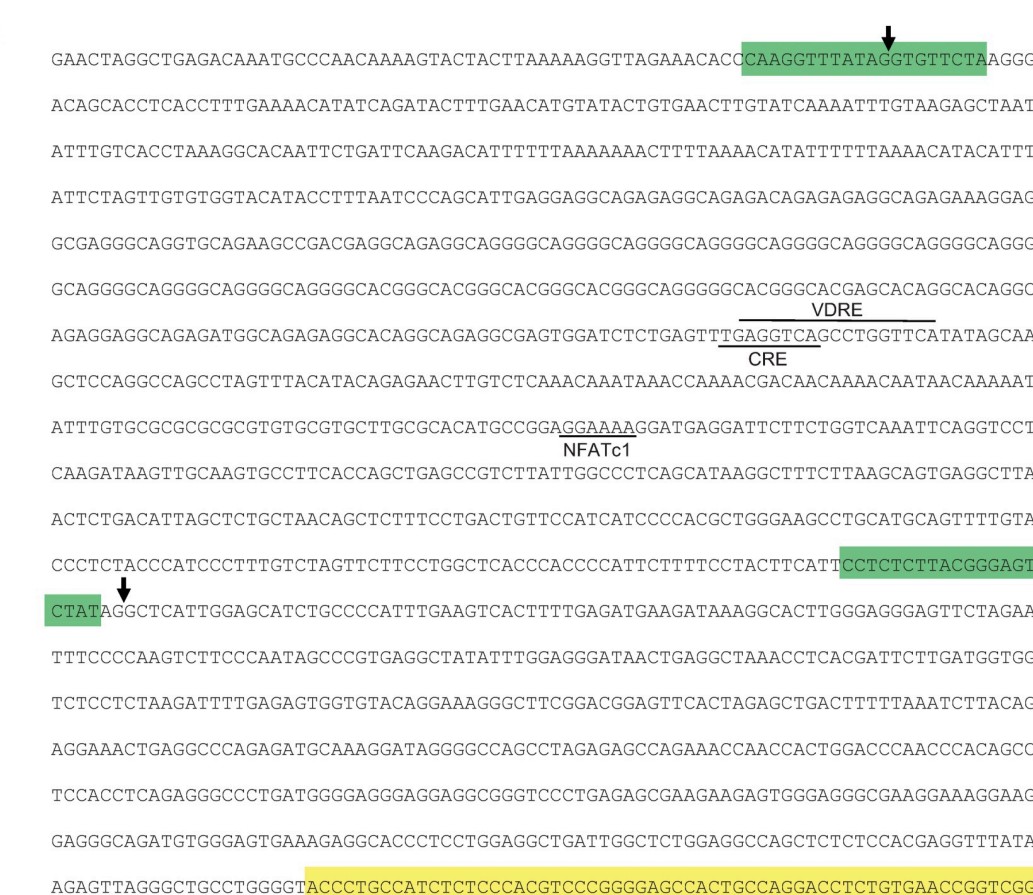

**Fig 1. Deletion of a potential proximal *Tnfsf11* enhancer from mice.** (**A**) Diagram showing sequence conservation between the murine and human *Tnfsf11* 5´-flanking region generated by the ECR Browser [20]. Numbers to the right indicate the percent homology between the murine and human sequence and numbers at the bottom indicate position in base pairs relative to the transcription start-site, which is designated +1. Regions upstream from the start-site that are at least 75 bp long and exhibit at least 70% similarity are indicated as red peaks. Yellow designates the 5´-untranslated region and blue designates the coding region in exon 1. The region deleted in this study is indicated by the bracket at the top of the figure. (**B**) Sequence of the proximal murine *Tnfsf11* 5´-flanking region and the beginning region of exon 1 (yellow highlight). The position of the sgRNAs used to delete the potential enhancer are highlighted in green and the arrows indicate the position of the cut sites, determined by sequencing. Putative transcription factor binding sites identified by previous studies are indicated by the horizontal lines.

cancellous bone volume did not differ between genotypes at either age, nor did femoral cortical thickness (**Fig 3A–3D**). Similarly, cancellous bone volume in the spine also did not differ between genotypes (**Fig 3E and 3F**). Taken together, the BMD and micro-CT results

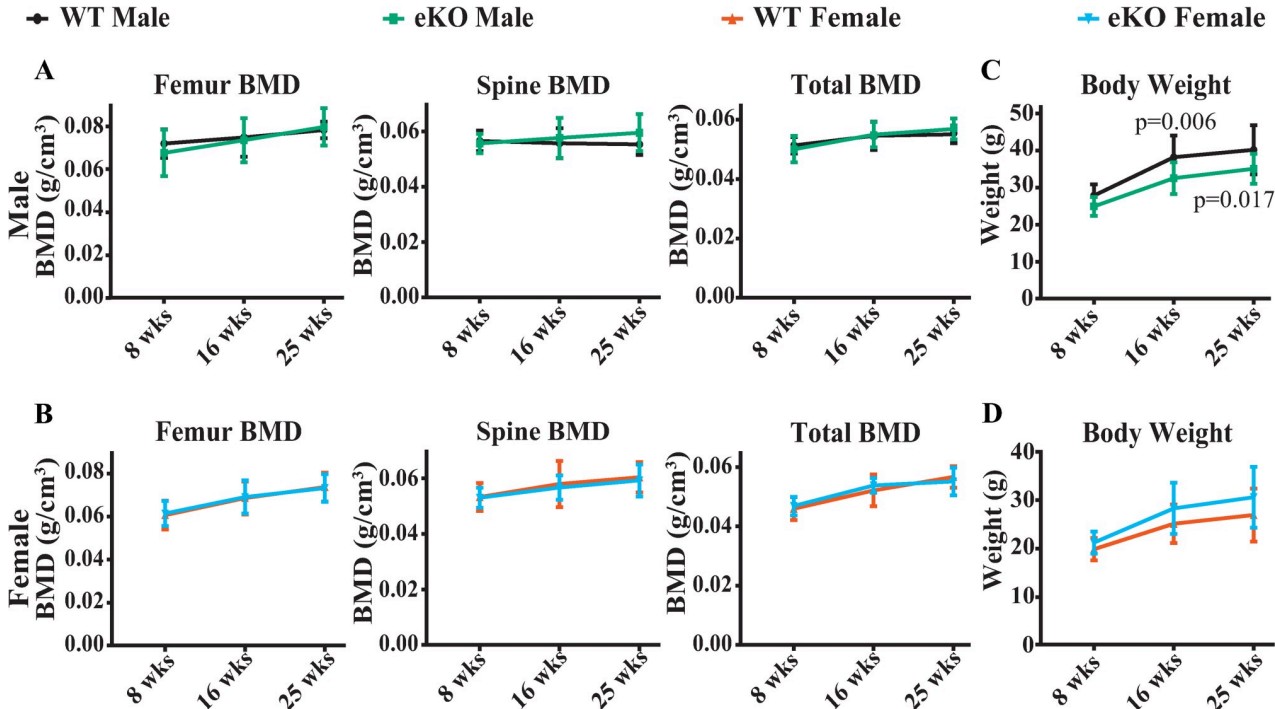

**Fig 2. Deletion of the potential enhancer does not alter BMD.** Serial analysis of BMD beginning at 8 weeks of age until 25 weeks of age in male (**A**) and female (**B**) wild type and homozygous eKO mice. Body weight of male (**C**) and female (**D**) mice determined at the time of BMD measurement. n = 11 to 18 per group, p values determined using 2-way ANOVA. All values are means ± s.d..

demonstrate that deletion of the -510 and -1413 region did not alter bone mass or architecture in the cancellous or cortical bone compartments during growth or adulthood. These results also suggest that osteoclast formation and bone resorption were unaffected by the deletion.

Since micro-CT analysis did not reveal any differences between WT and eKO mice at either age in either sex, we limited further analysis to female mice. While there were no differences in bone mass, deletion of the enhancer could still have a small impact on expression of *Tnfsf11* in bone or affect expression in other tissues. To test this, we quantified *Tnfsf11* transcripts in RNA extracted from fifth lumbar vertebrae (L5), tibial cortical bone, spleen, and thymus of WT and eKO mice at 6 months of age by quantitative PCR. We did not observe differences in *Tnfsf11* mRNA levels between genotypes in any of these tissues (**Fig 4A**), nor did we detect differences in circulating RANKL protein (**Fig 4B**). Consistent with no change in *Tnfsf11* mRNA in the spleen and thymus, the percentage of B and T lymphocytes in the bone marrow, as measured by CD19-positive and CD3-positive cells, respectively, was not altered by deletion of the putative enhancer (**Fig 4C**).

Lastly, as a stringent test for a possible role for the -510 and -1413 region in *Tnfsf11* expression, we examined the responsiveness of this gene to PTH or 1,25(OH)$_2$D$_3$ in bone marrow stromal cells isolated from eKO and WT littermates. PTH potently stimulated *Tnfsf11* mRNA abundance in cells from both genotypes but the maximal stimulation was reduced by approximately 40% in cultures from the eKO mice (**Fig 4D**). This finding was replicated in a second experiment performed in cells from mice of the opposite sex (**S3 Fig**). In contrast, the potent stimulation by 1,25(OH)$_2$D$_3$ was not consistently changed by deletion of the enhancer region (**Fig 4E**). To confirm that the bone marrow cultures from each genotype had reached similar stages of development, we measured expression of *Tnfrsf11b* and its response to the hormones.

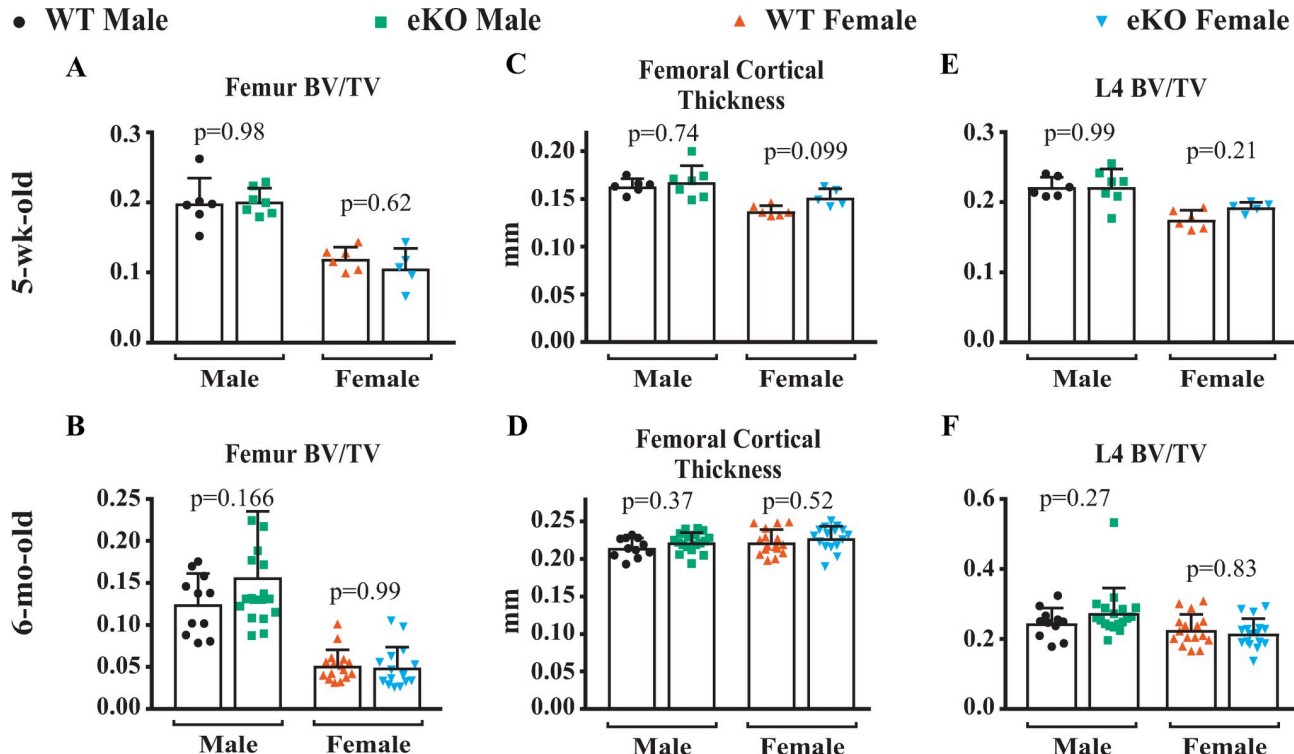

**Fig 3. Deletion of the potential enhancer does not alter cortical or cancellous bone mass.** Cancellous bone volume over total volume (BV/TV) of the femurs from male and female mice of the indicated genotypes measured at 5 weeks of age, n = 5–7 (**A**) and 6 months of age, n = 11–18 (**B**). Femoral cortical thickness in 5-week-old mice (**C**) and 6-month-old mice (**D**). Cancellous BV/TV of the L4 vertebrae 5-week-old mice (**E**) and 6-month-old mice (**F**). Values are means ± s.d..

Suppression of *Tnfrsf11b* by either hormone was unaffected by deletion of the enhancer region (**Fig 4F and 4G**).

## Discussion

Osteoclasts are essential during bone growth for removal of calcified cartilage beneath the growth plate, for modeling of bone shape, and for remodeling of bone matrix. Consequently, reductions in osteoclast formation or activity necessarily result in altered bone mass and structure. Specifically, reduced osteoclast number due to reduced *Tnfsf11* abundance leads to increased cancellous and cortical bone mass [3,21]. However, in the current study we did not observe any differences in bone mass or structure between eKO mice and WT littermates during growth or adulthood. We also did not detect any reductions in *Tnfsf11* mRNA in bones of mice lacking the -510 to -1413 region. Therefore, our results do not support a role for the -510 to -1413 region, and the putative transcription factor binding sites therein, in the control of *Tnfsf11* expression in the bones of growing or adult mice under normal physiological conditions.

As RANKL has functions in non-skeletal tissues [3,4,22], we also measured *Tnfsf11* mRNA in thymus and spleen and again observed no changes after deletion of the potential enhancer. Consistent with this, we did not observe any changes in B or T lymphocyte abundance in the bone marrow. Although the -510 to -1413 region does not contribute to *Tnfsf11* expression in bone or lymphocytes, it remains possible that this region is involved in *Tnfsf11* expression in cell types important for M cell, lymph gland, or mammary tissue development, which were not examined in this study. It is also possible that this region contributes to *Tnfsf11* expression in

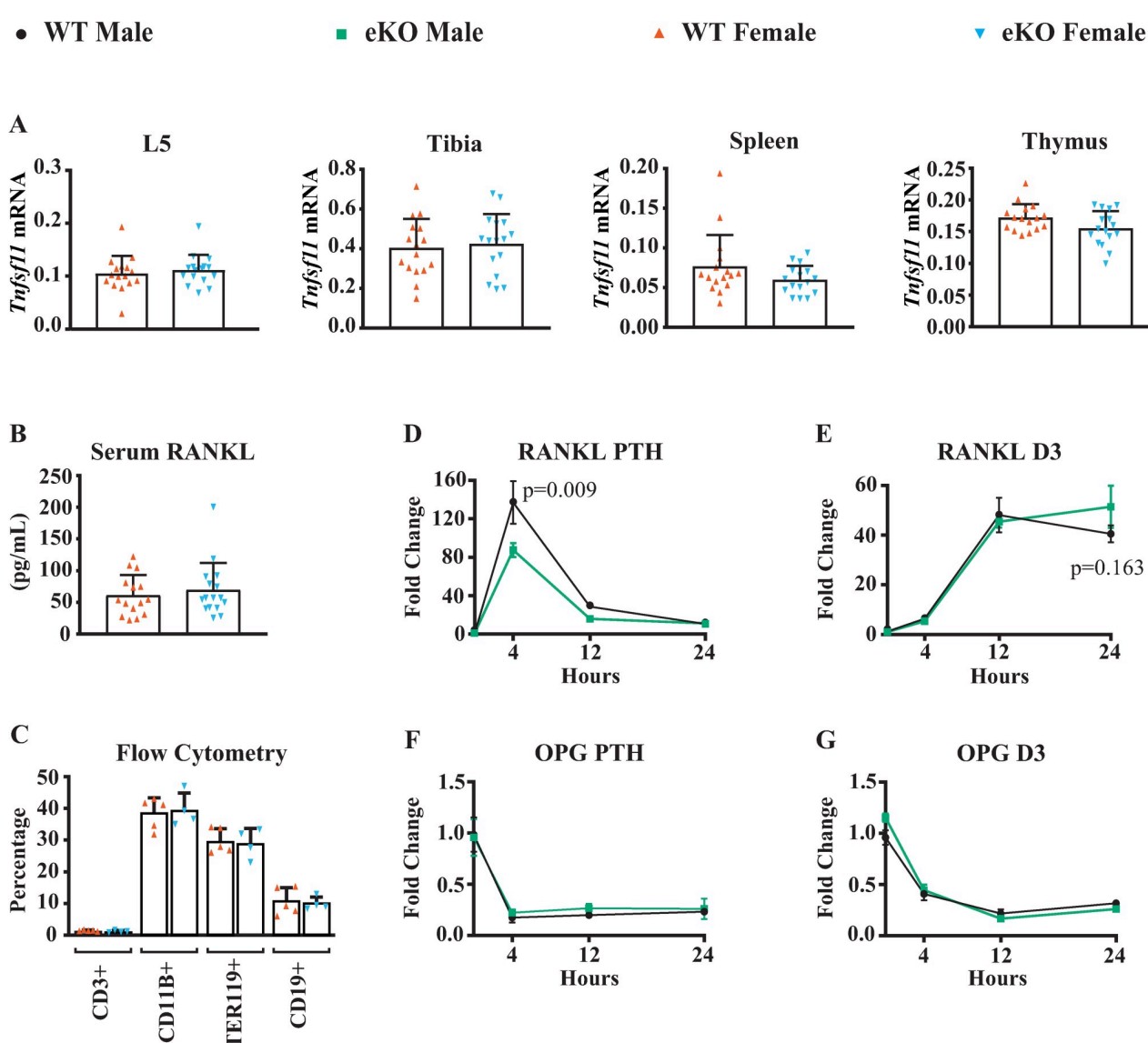

**Fig 4. Impact of enhancer deletion on *Tnfsf11* expression.** (**A**) *Tnfsf11* mRNA measured by Taqman RT-PCR in RNA prepared from L5 vertebrae, tibial cortical bone, spleen, and thymus of 6-month-old female mice of the indicated genotypes. n = 16–18 per group. (**B**) RANKL protein measured in the serum of 6-month-old female mice of the indicated genotypes. n = 15–16 per group. (**C**) Flow cytometry analysis of bone marrow hematopoietic cells from 6-month-old female mice of the indicated genotypes. n = 4–5 per group. Values are means ± s.d.. (**D-G**) *Tnfsf11* (RANKL) and *Tnfrsf11b* (OPG) mRNA measured by Taqman RT-PCR in RNA prepared from bone marrow stromal cell cultures from male mice of the indicated genotype treated with vehicle, $10^{-7}$ M PTH, or $10^{-8M}$ 1,25(OH)$_2$D$_3$ for the indicated times. Values are the mean fold-change, ± s.d., of triplicate wells, p values determined eKO versus WT cultures at the same time point by Repeated Measures Mixed Effects models.

bone under conditions that were also not examined here. For example, the finding that the response to PTH in vitro was blunted suggests that this region may be involved in the elevated *Tnfsf11* expression associated with states of PTH excess. However, it is important to note that PTH is a tonic stimulator of *Tnfsf11* under normal physiological conditions [23]. Thus, the unchanged abundance of *Tnfsf11* mRNA in bones of the eKO mice suggests that the enhancer is not required for response to physiologically normal levels of PTH, PTHrP, or 1,25(OH)$_2$D$_3$.

One limitation of our study was that the mice were not in a homogeneous genetic background. The majority of the animals used in our study were in a mixed C57BL/6-BalbC

background with a slightly higher contribution from C57BL/6 than BalbC. Nonetheless, our previous study of a different *Tnfsf11* enhancer mouse model, with only slightly less genetic heterogeneity, clearly revealed differences in bone mass and gene expression in mice lacking a more distal enhancer [7,9]. This, combined with the high number of animals per group in the current study, suggests that if the region between -510 and -1413 contributes to *Tnfsf11* expression, the contribution is small. The genetic heterogeneity may also have contributed to the slight difference in body weight observed at some ages in the male eKO mice.

Although we found no evidence that the -510 to -1413 region contributes to the control of *Tnfsf11* by 1,25$(OH)_2D_3$, previous studies identified a putative VDRE within this region [11,24]. It is important to note that these previous studies relied on transient transfection of promoter-reporter constructs into osteoblastic cell lines. This approach suffers from at least 2 major limitations. First, the DNA fragments used in such studies typically contain relatively short spans, typically less than 2–3 kb, of DNA upstream from the transcription start site. Thus, any regulatory activity observed using these constructs represents that action of cis-acting elements removed from their normal environment of flanking regulatory elements. Second, DNA introduced into cells by transient transfection tests the function of such constructs in an episomal state, which has been demonstrated to produce results that differ from those obtained after integration of reporter DNA into nuclear genomic DNA [25,26]. Thus, results from studies that rely exclusively on transient transfection of promoter-reporter constructs may not accurately reflect the regulation of the endogenous gene. In fact, the regulation of many genes established through analysis of sites located near their promoters using cultured cell-based transfection studies has not been confirmed through ChIP-seq analysis and other methods in vivo. The results of the present study highlight the necessity to test the function of transcriptional regulatory elements in the context of their native environment, which has been greatly facilitated by the development of CRISPR-Cas-based gene editing.

## Supporting information

**S1 Fig. Sequencing of the mutant allele (eKO).** (**A**) PCR products from the genotyping PCR described in the Material and methods section were fractionated on an agarose gel, stained with ethidium bromide, and imaged. The size of the eKO product is 261 bp and that of the WT product is 185 bp. Representative results are shown for a homozygous eKO mouse, heterozygous eKO mice, and a WT mouse. (**B**) The sequence of the eKO allele was determined by TA-cloning of the 261 bp PCR product and sequencing. The relevant portion of the sequencing result is shown aligned to the sequence of the WT allele. 855 bp of the WT allele were ommitted from the figure to reduce the size of the sequence. The PAMs for each of the sgRNAs used to create the deletion are highlighted in green.
(TIF)

**S2 Fig. Body composition and femur length.** Serial analysis of lean and fat mass using Piximus-derived fat percentage and total tissue mass data beginning at 8 weeks of age until 25 weeks of age in male (**A**) and female (**B**) wild type and homozygous eKO mice. n = 11 to 18 per group, p values determined using 2-way ANOVA. (**C-D**) Femur length of 5-week-old and 6-month-old mice using right femurs measured with calipers. n = 3 to 18 per group, p values determined using 2-way ANOVA. Values are means ± s.d..
(TIF)

**S3 Fig. *Tnfsf11* expression in cultured bone marrow stromal cells.** (**A-D**) *Tnfsf11* (RANKL) and *Tnfrsf11b* (OPG) mRNA measured by Taqman RT-PCR in RNA prepared from bone marrow stromal cell cultures from female mice of the indicated genotype treated with vehicle,

$10^{-7}$ M PTH, or $10^{-8M}$ 1,25(OH)$_2$D$_3$ for the indicated times. Values are the mean fold-change, ± s.d., of triplicate wells; p values determined by comparing eKO versus WT values at the same time point by Repeated Measures Mixed Effects models.
(TIF)

**S1 Raw images.**
(TIF)

## Acknowledgments

We thank P.E. Baltz, C. Bustamante-Gomez, and the staff of the UAMS Department of Laboratory Animal Medicine for technical support.

## Author Contributions

**Conceptualization:** Ryan S. MacLeod, Mark B. Meyer, J. Wesley Pike, Charles A. O'Brien.

**Formal analysis:** Ryan S. MacLeod, Jeff D. Thostenson, Charles A. O'Brien.

**Funding acquisition:** Charles A. O'Brien.

**Investigation:** Ryan S. MacLeod, Mark B. Meyer, Jinhu Xiong, Keisha M. Cawley, Yu Liu, Melda Onal, Nancy A. Benkusky.

**Methodology:** Ryan S. MacLeod, Mark B. Meyer, J. Wesley Pike, Charles A. O'Brien.

**Supervision:** Charles A. O'Brien.

**Writing – original draft:** Ryan S. MacLeod, Charles A. O'Brien.

**Writing – review & editing:** Ryan S. MacLeod, Mark B. Meyer, Jinhu Xiong, Keisha M. Cawley, Yu Liu, Melda Onal, Nancy A. Benkusky, Jeff D. Thostenson, J. Wesley Pike, Charles A. O'Brien.

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
