## [Decision Letter · Decision Letter 0]

22 Feb 2021

PONE-D-21-02120

Deletion of a putative promoter-proximal Tnfsf11 regulatory region in mice does not alter bone mass or Tnfsf11 expression in vivo

PLOS ONE

Dear Dr. O'Brien,

Thank you for submitting your manuscript to PLOS ONE. After careful consideration, we feel that it has merit but does not fully meet PLOS ONE’s publication criteria as it currently stands. Therefore, we invite you to submit a revised version of the manuscript that addresses the points raised during the review process.

The reviewers noted a few important gaps in your MS.  Most important is to document success of the targeting strategy.  In addition, more details about growth and body composition are sought.  While thew overall decision is classified as "major revision" because of their importance to the story, making the requested changes should be relatively straightforward.

We look forward to receiving your revised manuscript.

Kind regards,

Robert Daniel Blank, MD, PhD

Academic Editor

PLOS ONE

"I have read the journal's policy and the authors of this manuscript have the following competing interests: CAO owns stock in Radius Health, Inc.."

Reviewers' comments:

Reviewer's Responses to Questions

**Comments to the Author**

1. Is the manuscript technically sound, and do the data support the conclusions?

Reviewer #1: Partly

Reviewer #2: Yes

2. Has the statistical analysis been performed appropriately and rigorously? 

Reviewer #1: Yes

Reviewer #2: Yes

3. Have the authors made all data underlying the findings in their manuscript fully available?

Reviewer #1: Yes

Reviewer #2: Yes

4. Is the manuscript presented in an intelligible fashion and written in standard English?

Reviewer #1: Yes

Reviewer #2: Yes

5. Review Comments to the Author

Reviewer #1: This work from MacLeod et al indicates that a region upstream of the RANKL (Tnfsf11) gene does not play an essential role in regulating bone structure, although it may be partially required for normal response of bone marrow-derived stromal cells to PTH stimulation of that gene. The data presented is clear, and limitations are clearly out lined in the discussion, but the rationale of the study and the success of the targeting strategy are not clear.

Major concerns:

1. No data is presented to show that the targeting strategy was effective. This is a major shortcoming as it calls all conclusions into question. Please show that recombination was successful.

2. Figure 1 is not very clear and the statement of hypothesis is a bit muddy in some instances. For those who are interested in RANKL regulation, but are not conversant with all the regions, it becomes difficult to follow. It is really important to show in Figure 1 how the targeted region aligns with the previously described enhancers at 22kb, and 75-76kb. You that you are targeting a region lacking "hormonal responsiveness" (lines 207-211), but this is also the region that contains the CRE and VDRE? Please clarify.

3. RANKL is also highly responsive to IL-6, and this group has published much on that issue. Where is the putative IL-6 response site (STAT binding element) compared to the region deleted?

4. The in vitro studies showing downregulation of the PTH effect on RANKL is based on a single cell preparation. This is not adequate and should either be removed or repeated so it is clear that the result was reproducible.

Minor concerns:

1. Methods: please provide more detail on the regions measured by micro-CT.

2. Please indicate in all figure legends whether error bars are SD or SEM.

3. The title for the legend of Figure 4 is a bit misleading as this also contains in vitro data.

4. Since PTHrP is required for normal osteoclast formation (see Miao et al 2005) and acts through the same receptor as PTH, you should include PTHrP in your statement on line 316 that the enhancer is not required for physiological action.

5. For full data transparency, you should show individual data points for Figure 2, and Figure 4C-G.

Reviewer #2: An enhancer near to the transcriptional start site has been reported for the Tnfsf11 gene, that codes for RANKL. In this study, the authors generate a transgenic mouse where in this enhancer has been eliminated completely. They evaluate the impact of the loss of this enhancer in vivo and in vitro and conclude that outside of high PTH conditions, this enhancer does not appear to play a role in bone. This is a beautifully written and clear paper. The experiments conducted are appropriate to address the role of this enhancer in Tnfsf11 and the results are presented in a clear fashion. I do have some recommendations and a couple of questions on issues that were not addressed in this paper.

1. Figure 2. Please provide body length to help interpret the body weigh issue. Are the knockout mice leaner, lacking mean mass or just over all smaller? The data on lean and fat mass should already be available from the DXA files.

2. Figure 3. Related to the comment above, please provide femur lengths. Given the nearly significant result for cortical thickness for the 5 week old females, it would be ideal to present cortical area and marrow area as well. Please place the P values for the within gender comparisons for this figure for transparency purposes.

3. Figure 4E. Arguably you do have a response to Vit D3. It is hard to tell what error bar belongs to what point. Please provide the P value for the 24 hour time point. The statements in the discussion regarding Vit D and this enhancer need to be tempered and not be such an absolute. With three samples for this assay, what is your power of detection for a difference here? The error bars are certainly wider for the KO than for the WT.

4. The issue of body weight is not mentioned in the discussion. Do you have any explanation for the reduction in body weight? As this mouse is not completely without a phenotype, the discussion needs to be expanded to reflect this.

6. PLOS authors have the option to publish the peer review history of their article (what does this mean?). If published, this will include your full peer review and any attached files.

Reviewer #1: No

Reviewer #2: No

---

## [Author Response · Author response to Decision Letter 0]

1 Apr 2021

Response to Reviewers

Reviewer 1.

Major concerns:

1. No data is presented to show that the targeting strategy was effective. This is a major shortcoming as it calls all conclusions into question. Please show that recombination was successful.

We regret that we did not more carefully explain how we confirmed the sequence of the deleted allele. We now state clearly that a PCR product containing the deletion junction was cloned and sequenced. An example of the PCR and an alignment of the sequence of the deleted allele with the WT allele is presented in new S1 Fig. 

2. Figure 1 is not very clear and the statement of hypothesis is a bit muddy in some instances. For those who are interested in RANKL regulation, but are not conversant with all the regions, it becomes difficult to follow. It is really important to show in Figure 1 how the targeted region aligns with the previously described enhancers at 22kb, and 75-76kb. You that you are targeting a region lacking "hormonal responsiveness" (lines 207-211), but this is also the region that contains the CRE and VDRE? Please clarify.

Our previous in vitro studies showed that the first 2000 bp of DNA upstream from the TSS were unable to confer responsiveness to either PTH or 1,25(OH)2D3 upon stable transfection of promoter-reporter constructs. In contrast, others have shown that similar constructs do display responsiveness to PTH and 1,25(OH)2D3, as well as isoproterenol. Thus, our previous studies suggest that this region is not important for Tnfsf11 responsiveness to these hormones while those of other groups suggest that it is important.

In an attempt to resolve these conflicting results, we deleted a fragment containing the putative response elements identified by the other groups from the endogenous murine Tnfsf11 gene and examined the impact on bone mass and Tnfsf11 gene expression. We have edited the text of the Results section to include more of the information that we presented in the Introduction section in an effort to clarify the goal of the present study.

Due to the large distance of the 22 kb and 75-76 kb enhancers from the TSS, their inclusion in the alignment in Fig 1 is not practical. Since no aspect of the current study deals with these distant enhancers, it may in fact be confusing to include them in the diagram in Fig 1, which deals exclusively with the proximal promoter region that we study in the current work.

3. RANKL is also highly responsive to IL-6, and this group has published much on that issue. Where is the putative IL-6 response site (STAT binding element) compared to the region deleted?

IL-6 is a member of the gp130 family of cytokines and we have shown that the enhancer mediating the bulk of the response to these cytokines is located 88 kb upstream from the TSS (Endocrinology 157:482, 2016). However, there are no conflicting results regarding the location of the response elements to gp130 family cytokines. Therefore, we think that discussion of their location in the present work will likely be distracting.

4. The in vitro studies showing downregulation of the PTH effect on RANKL is based on a single cell preparation. This is not adequate and should either be removed or repeated so it is clear that the result was reproducible.

We thank the reviewer for making this point and have included a second experiment performed using cells from female mice. The first experiment used cells from male mice. The results of both experiments are very similar, showing a mild reduction in the response to PTH at 4 hr of treatment. The second experiment showed a mild increase in the response to 1,25(OH)2D3 in cells from the eKO mice at the 24 hr time-point. Since this did not occur in the first experiment, this latter result likely represents experimental variation. We have included the new results in S3 Fig.

Minor concerns:

1. Methods: please provide more detail on the regions measured by micro-CT.

We have added details for both the femur and vertebral measurements.

2. Please indicate in all figure legends whether error bars are SD or SEM.

We apologize for this omission and this information is now included in the figure legends.

3. The title for the legend of Figure 4 is a bit misleading as this also contains in vitro data.

We agree and have altered the title for this figure legend.

4. Since PTHrP is required for normal osteoclast formation (see Miao et al 2005) and acts through the same receptor as PTH, you should include PTHrP in your statement on line 316 that the enhancer is not required for physiological action.

We have included PTHrP in this conclusion.

5. For full data transparency, you should show individual data points for Figure 2, and Figure 4C-G.

We have included individual data points for Fig 4C. However, when we did so for the line graphs in Fig 2 and Fig 4, the points representing the means were obscured and the graphs become essentially unintelligible. Thus, for these later graphs, we think it is clearer not to include individual data points.

Reviewer #2

1. Figure 2. Please provide body length to help interpret the body weigh issue. Are the knockout mice leaner, lacking mean mass or just over all smaller? The data on lean and fat mass should already be available from the DXA files.

Unfortunately we did not collect body length measurements on these mice. Because the mice are not uniformly extended in the DXA scans, we could not use the scans for such measurements. As suggested by the reviewer, we have included the lean and fat mass measurements from the DXA scans in a new figure, S2 Fig. Overall, the male eKO mice are slightly leaner and have slightly less fat. 

2. Figure 3. Related to the comment above, please provide femur lengths. Given the nearly significant result for cortical thickness for the 5 week old females, it would be ideal to present cortical area and marrow area as well. Please place the P values for the within gender comparisons for this figure for transparency purposes.

The femur lengths did not differ by genotype and are now included in the new S2 Fig. As suggested, we now include the within gender P values in Fig 3.

3. Figure 4E. Arguably you do have a response to Vit D3. It is hard to tell what error bar belongs to what point. Please provide the P value for the 24 hour time point. The statements in the discussion regarding Vit D and this enhancer need to be tempered and not be such an absolute. With three samples for this assay, what is your power of detection for a difference here? The error bars are certainly wider for the KO than for the WT.

We now include the P value (p = 0.163) for the 24 hr time-point. Based on the lack of significance here, and our finding that in a second experiment, the response to 1,25(OH)2D3 was increased in cells from the eKO mice at 24 hr, we conclude that there is not a consistent change in the response to this hormone. 

4. The issue of body weight is not mentioned in the discussion. Do you have any explanation for the reduction in body weight? As this mouse is not completely without a phenotype, the discussion needs to be expanded to reflect this.

We think the mostly likely explanation for the slightly lower body weight of the male eKO mice is genetic heterogeneity. The female eKO mice appear to be slightly heavier than controls, but this does not reach statistical significance. The overall effect size is similar in both sexes, just in the opposite direction. Based on this, and the fact that there is no evidence in any previous Tnfsf11 loss-of-function study that this cytokine plays a role in the control of body weight, we think that the mostly likely explanation for these effects is that our animals are not in a pure genetic background. We have now mentioned this possibility in the Discussion section.

---

## [Decision Letter · Decision Letter 1]

19 Apr 2021

Deletion of a putative promoter-proximal Tnfsf11 regulatory region in mice does not alter bone mass or Tnfsf11 expression in vivo

PONE-D-21-02120R1

Dear Dr. O'Brien,

We’re pleased to inform you that your manuscript has been judged scientifically suitable for publication and will be formally accepted for publication once it meets all outstanding technical requirements.

Kind regards,

Robert Daniel Blank, MD, PhD

Academic Editor

PLOS ONE

Additional Editor Comments (optional):

Reviewers' comments:

Reviewer's Responses to Questions

**Comments to the Author**

1. If the authors have adequately addressed your comments raised in a previous round of review and you feel that this manuscript is now acceptable for publication, you may indicate that here to bypass the “Comments to the Author” section, enter your conflict of interest statement in the “Confidential to Editor” section, and submit your "Accept" recommendation.

Reviewer #1: All comments have been addressed

Reviewer #2: All comments have been addressed

2. Is the manuscript technically sound, and do the data support the conclusions?

Reviewer #1: Yes

Reviewer #2: Yes

3. Has the statistical analysis been performed appropriately and rigorously? 

Reviewer #1: Yes

Reviewer #2: Yes

4. Have the authors made all data underlying the findings in their manuscript fully available?

Reviewer #1: Yes

Reviewer #2: Yes

5. Is the manuscript presented in an intelligible fashion and written in standard English?

Reviewer #1: Yes

Reviewer #2: Yes

6. Review Comments to the Author

Reviewer #1: (No Response)

Reviewer #2: I would like to thank the authors for their consideration of all comments made by both reviewers for this manuscript.

7. PLOS authors have the option to publish the peer review history of their article (what does this mean?). If published, this will include your full peer review and any attached files.

Reviewer #1: No

Reviewer #2: No

---

## [Editor Report · Acceptance letter]

30 Apr 2021

PONE-D-21-02120R1 

Deletion of a putative promoter-proximal *Tnfsf11* regulatory region in mice does not alter bone mass or *Tnfsf11* expression in vivo 

Dear Dr. O'Brien:

I'm pleased to inform you that your manuscript has been deemed suitable for publication in PLOS ONE. Congratulations! Your manuscript is now with our production department. 

Kind regards, 

on behalf of

Professor Robert Daniel Blank 

Academic Editor

PLOS ONE